# (Pre-)training Dynamics: Scaling Generalization with First-Order Logic

## Abstract

Transformer-based models have demonstrated a remarkable capacity for learning complex nonlinear relationships. While previous research on generalization dynamics has primarily focused on small transformers (1-2 layers) and simple tasks like XOR and modular addition, we extend this investigation to larger models with 125M parameters, trained on a more sophisticated first-order logic (FOL) task. We introduce a novel FOL dataset that allows us to systematically explore generalization across varying levels of complexity. Our analysis of the pretraining dynamics reveals a series of distinct phase transitions corresponding to the hierarchical generalization of increasingly complex operators and rule sets within the FOL framework. Our task and model establish a testbed for investigating pretraining dynamics at scale, offering a foundation for future research on the learning trajectories of advanced AI systems.

## 1 Introduction

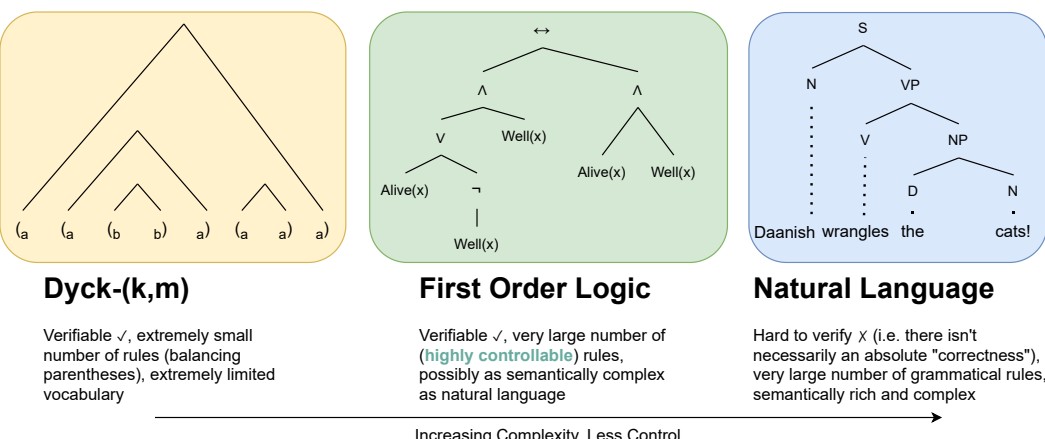

Figure 1: First order logic (FOL) problem in the context of language modeling complexity.

Transformers achieve state-of-the-art performance across a wide range of tasks, but the mechanisms that enable their effective generalization are not yet fully understood. Current interpretability methods primarily focus on identifying linearly separable features, which overlook the complex, nonlinear interactions that transformers exploit, such as XOR-like feature combinations that seem to be essential for generalized learning, and that have been observed empirically (Marks, 2023).

A striking example of generalization in training dynamics is grokking, first discovered with overfitting transformers on algorithmic datasets (Power et al., 2022). Subsequently, grokking has been extensively studied with various algorithmic problems such as arithmatic, modular addition, and XOR (Nanda et al., 2023), but Liu et al. (2022b) suggests that grokking can be induced with more realistic data. With intuitions gained from toy model settings such as better representation learned by the embeddings or higher initial weights, it suggests that grokking may occur with natural language as

well. Current research on grokking, however, remains quite distant from being applicable to natural language.

As highlighted in previous studies on grokking and generalization, generalization is usually tested with out-of-distribution data relative to the training set. This poses a challenge in the context of natural language, where distinguishing between between various categorizations of such unstructured data becomes difficult. For instance, the distinction between "reasoning" and "non-reasoning" text can be ambiguous. Consequently, research on grokking and generalization often employs algorithmic datasets, where the distinction between in-domain and out-of-domain data is clear, such as modular addition of three-digit integers versus five-digit integers. However, these algorithmic datasets often lack complexity, meaning they do not require large models for training. Furthermore, they do not adequately resemble natural language, making it difficult to draw parallels with the training of realistic LLMs, which are trained on highly diverse and unstructured natural language data. So, to scale up generalization studies, we must also scale up the problems too.

To that aim, we consider the task of learning first order logic (FOL). FOL combines various operators and parenthetical expressions to mark phrases and predicates in a way that resembles natural language. If we consider the spectrum of complexity with respect to natural language, we can situate FOL as shown in Figure 1. On the simpler end, Dyck languages, consisting of parenthetical closures, share a basic structural syntax of hierarchy similar to natural language. Due to this structural similarity, it has been extensively studied in the context of hierarchical learning in transformers (Hewitt et al., 2020; Murty et al., 2023; Yao et al., 2021; Manning et al., 2020), but it remains too abstract to legibly compare to natural language.

In terms of grammar, FOL is even closer to natural language, as it can express more intricate grammatical rules, including negation ($\neg$) and conjunctions ($\wedge$, $\vee$). FOL also shares structural similarities beyond simple rules, such as the composition of information within phrases demarcated with parentheses much like Dyck languages, as well as reasoning structures. Additionally, FOL incorporates semantic identifiers in its predicates, such as $Eats(x)$ or $HitchhikesToTheGalaxy(x)$, adding significant semantic complexity. While it cannot fully capture the unstructured nuances of natural language, FOL represents a subset of it. FOL stands as a step closer to natural language compared to other simplistic algorithmic tasks. One of its most notable advantages is that, despite its ability to represent complex concepts and even aspects of natural language, it remains controllable. FOL statements can be definitively verified as either correct or incorrect. This semi-algorithmic nature provides a unique and rare opportunity to quantify data complexity that can be scaled up or down as needed.

**In this work,** We explore the pretraining dynamics of transformers in a much larger and more complex setting compared to the shallow 1-2 layer transformers previously analyzed in grokking studies. To move beyond the simple algorithmic tasks commonly used in grokking models, we introduce a more challenging task: learning first-order logic (FOL). As this task is semi-algorithmic, it allows for greater control over the complexity of the dataset while aligning closer with natural language. This approach will enhance our understanding of the pretraining process of LLMs unstructured language data. We present a novel, pretraining-scale dataset based on FOL, specifically designed for this investigation. Through empirical analysis, we examine the generalization patterns that arise at this larger scale and complexity. Our results show that hierarchical generalization follows a staircase-like progression with distinct phases. Moreover, by analyzing the trajectories of operators and logical rules acquired during training, we gain deeper insights into the mechanisms driving each phase and how they contribute to the overall learning process.

Understanding the pretraining process is crucial, but it often remains obscure due to the vast size and complexity of the models. To build a tractable system, gaining insights into their learning process is essential. We address this by providing an effective testbed for exploring pretraining dynamics, to scale up future work in generalization research.

## 2 EXPERIMENTAL SETUP AND OVERVIEW

We begin by generating a synthetic pretraining corpus of FOL as detailed in Section 2.1.[1] This synthetic FOL dataset has syntactically simpler and controllable structures akin to algorithmic tasks, but retains the semantic richness of natural language. Table 1 provides some example data to demonstrate this. We then pretrain GPT-2-small implementation (Radford et al., 2019) on the FOL corpus. Specifically, we use a modified implementation of Karpathy's nanoGPT (Karpathy, 2024). Finally, we examine the resulting learning curves by different subsets of data, both in-domain and out-of-domain. We also examine the granular trajectories with particular operators of FOL and rule sets.

### 2.1 FOL CORPUS: PRETRAINING DATASET GENERATION

We crafted a synthetic pretraining dataset[2] with various LLMs and Sympy(Meurer et al., 2017), a python library for symbolic expressions.[3] We use Sympy for syntactic correctness of our random expressions, and we used LLMs for generating semantically varied expressions. The LLMs used for generating the logical expressions are much larger than a smaller model we are training. To train a GPT-2-small size model with 125M parameters, we estimated that we need to generate around 2.5B tokens as suggested by Hoffmann et al. (2022). Some examples of the FOL corpus are shown in Table 1.

| FOL Type | Example |
|---|---|
| Modus Tollens | $\forall x \text{AttendingParty}(x) \rightarrow \text{ExpectedFormalAttire}(x),$ 
 $\neg\text{ExpectedFormalAttire}(yoona) \rightarrow \neg\text{AttendingParty}(yoona)$ |
| Disjunctive Syllogism | $\forall x((\text{WatchMovie}(x) \vee \text{PlayGame}(x))),$ 
 $\neg\text{WatchMovie}(nadia) \rightarrow \text{PlayGame}(nadia)$ |
| Elimination (E11) | $\neg\text{Funny}(gerald) \vee \text{Funny}(gerald) \rightarrow \text{True}$ |
| Complex (C21) | $(\text{Symptoms}(x) \rightarrow (\text{GetsDiagnosis}(x) \vee \text{AccessesOptions}(x))),$ 
 $(\text{FollowsHealthGuidelines}(x) \rightarrow \text{Wellbeing}(x)),$ 
 $(\text{Symptoms}(x) \vee \text{FollowsHealthGuidelines}(x)) \rightarrow$ 
 $(\text{GetsDiagnosis}(x) \vee (\text{AccessesOptions}(x) \vee \text{Wellbeing}(x)))$ |
| Randomly Generated *And Correct* Expression | $((\text{CosmicBackgroundRadiation}(x) \wedge \text{FormationOfStars}(x))$ 
 $\vee\neg\text{CosmicBackgroundRadiation}(x) \vee \neg\text{FormationOfStars}(x))$ 
 $\leftrightarrow (\text{True})$ |

Table 1: Examples of First Order Logic (FOL) pretraining data and their categories. The explanations for each FOL categories are detailed in the Appendix A, B, and C.

To illustrate how FOL can represent logic, we take a look at an example of the *Eliminations* Rule (E11) given in Table 1. We can translate it to natural language as,

$$\{gerald \text{ is not } (\neg) \text{ funny}\} \text{ or } (\vee) \{gerald \text{ is funny}\}$$
$$\text{implies } (\rightarrow) \text{ True}.$$

Given a True or False function, Funny$(x)$, this statement has to be True. There are multiple such basic properties and inference rules that make up the "grammar" of FOL, as outlined in Appendix A. Particularly, elimination rules as shown in Appendix B are useful for simplifying FOL expressions and determining equivalences.

In order to teach FOL to a small scale LLM, we mass generate many such examples using much larger LLMs. We primarily used GPT models (GPT-3.5-turbo, GPT-4-turbo, GPT-4o, and GPT-4-mini) (Achiam et al., 2023) and Reka models (Core and Flash) (Ormazabal et al., 2024) to generate by providing symbolic FOL rules and in-context examples in the prompts. The in-context examples were provided from the existing high quality human annotated datasets, Folio (Han et al., 2022) and LogicBench (Parmar et al., 2024). In addition to the basic properties (Table 3), inferencing rules

---

[1]The models and all checkpoints will be released upon publication.
[2]The datasets will be released upon publication.
[3]The code and prompts used for generating the dataset will be released upon publication.

(Table 2), and elimination rules (Table 4), we can also craft more complex FOL expressions that specifically combines a combination of annotated FOL properties and inferencing rules, as shown in Table 5. To generate more unique and correct FOL rules at scale, we used Sympy to mass generate 400-500K unique rules of 1-8 variables, depth 1-4 and 1-4 sub-expressions per depth. Sympy relies on graphical representation of FOL operations, and therefore, it can guarantee correctness of generated expression as well as its simplifications. Around 70% of the training data consists of the randomly generated and guaranteed correct expressions and their equivalent simplifications. The full breakdown of the training data is summarized in Table 6.

## 2.2 Designing the Test Sets

For our test data, we withhold a subset of the generated data as our validation set. Existing human curated datasets such as Folio and LogicBench were used as another "human validation set." Furthermore, in order to truly test generalization, we attempt to create test examples that the model has never seen before. Since our model has only seen first-order logic, we use Dyck-$(k, m)$ languages as our generalization set, where $k$ = number of parenthesis types and $m$ = maximum depths of parenthetical expressions. Using the setup from Hewitt et al. (2020), we generated dyck languages of varying depths and vocabulary with finite-state automata. We hope to create an analogy for controllable complexity of vocabulary (controllable semantic complexity) and controllable syntactical complexity (number of nesting that occurs). Furthermore, we created complex chains of rules that combine varying numbers of basic inference properties as summarized in Appendix C. We then include some of the rules (C2, C3, C4, C5, C7, C8, C10, C11, C13, C14, C17, C20, C21, C23) in our pretraining, and withheld some (C1, C6, C9, C12, C15, C18, C22) for another test of generalization. Sympy was used to generate 400-500K syntactical rules, it is highly unlikely to have generated our exact sets of complex rules, with the same variables, predicates, and orders of operations. While the complex rule sets demonstrate varying levels of complexity by combining differing numbers of basic inference properties, the rule set represents a limited number of syntactic variety.

## 2.3 Pretrain an LLM on FOL corpus

We train nanoGPT with 125M parameters, Karpathy (2024)'s implementation of GPT-2-small, with 12 layers and 12 heads per layer. We pretrain from scratch on our custom FOL corpus. We used an embedding size of 768 and block size of 1024 tokens and a micro-batch size of 12, with gradient accumulation steps set to 40 ($5 \times 8$) to simulate a larger effective batch size. No dropout was applied during pre-training, and the AdamW optimizer is used with a learning rate of $1 \times 10^{-4}$, weight decay of 0.1, and gradient clipping at 1.0. Learning rate included a warm-up phase over 1000 iterations, with a decaying schedule until a minimum learning rate of $1 \times 10^{-5}$, over a total of 10,000 iterations. We trained on 4 NVIDIA RTX A6000 for 62.8 hours.

## 3 Results

### 3.1 Learning Curves

The training curves for pretraining on the FOL corpus is shown in Figure 2. We see that there are multiple phase transitions, captured by various test sets including our human annotated datasets, and withheld complex chains of first order logic simplification derivations. Generally, we see that the validation and human annotated validation sets follow similar trajectories as the training curve. To assess generalization, we used Dyck languages with varying depths and types of parentheses for validation, since we assume they possess a significantly different data distribution compared to FOL and therefore appropriately out-of-domain. As shown in Figure 1, Dyck languages consist of parentheses closures, making them an effective testbed for evaluating whether the model understands syntactical hierarchies. The Dyck languages validation curves reveal hierarchical generalization occurring in staircase-like phases. We label these regions by the phases of Dyck language losses, as shaded and labeled in Figure 2.

We examine learning dynamics at various hierarchies with Dyck languages as shown in Figure 3. Interestingly, we see that there might be multiple phases not captured by our test sets. Moreover,

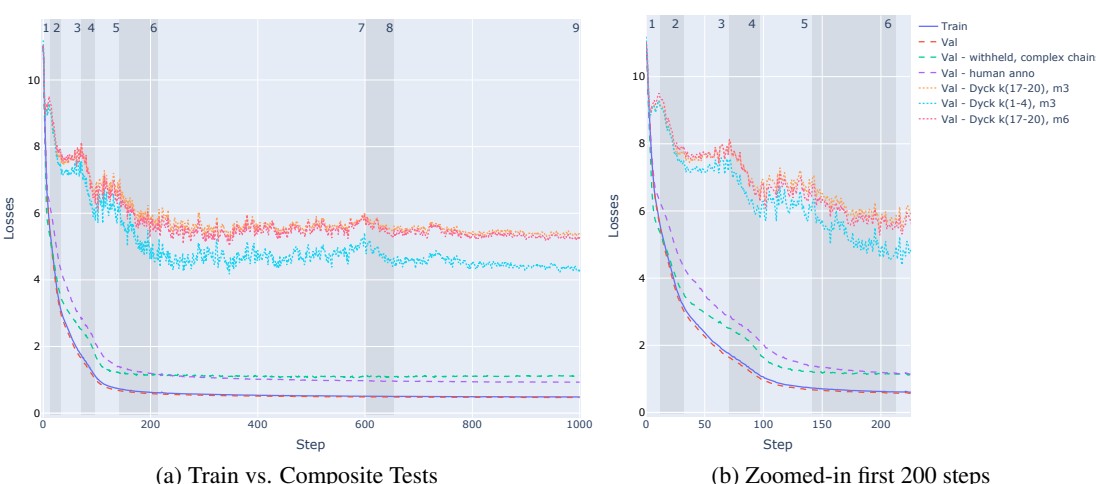

(a) Train vs. Composite Tests

(b) Zoomed-in first 200 steps

Figure 2: Training and Validation Curves for FOL pretraining

upon zooming into phase 4 region in Figure 3b, we see that there is an inflection point at which the losses for shallower expressions increase past higher depth expressions. After this inflection point, the model exhibits higher loss for lower depth expressions than higher depth expressions.

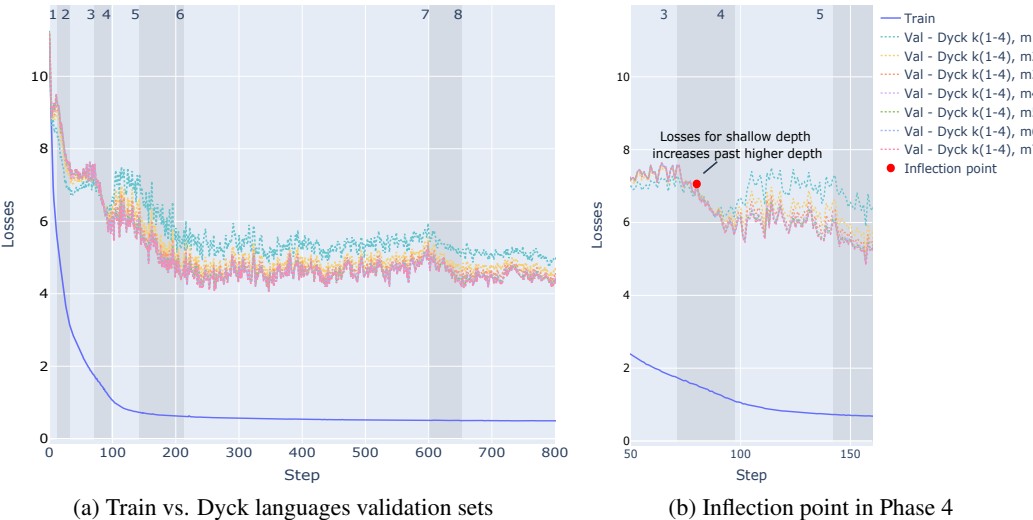

(a) Train vs. Dyck languages validation sets

(b) Inflection point in Phase 4

Figure 3: Dyck languages of vocab 1-4 and varying depths

**Normalized Per-Token Loss of Dyck Languages**   We hypothesize that lower depth expressions in phase 4 and beyond exhibit higher loss because the model has fewer previous tokens to condition on, resulting in worse predictive performance. This is exacerbated by the fact that our Dyck language test sets have a token distribution that is quite different from that of our training data, as they only utilize a subset of tokens—specifically, the parentheses. We suspect that the longer expressions may help the model narrow its distribution to the valid tokens even if the model has not learn the underlying syntactic rules.

To reduce this bias, we look at a normalized per-token loss that captures the negative log-likelihood placed by the model on the correct next token when restricted to the set of valid tokens for that test set. We compute this by setting the logits of invalid tokens to -$\infty$ before loss computation. Because of the use of softmax in logit normalization, setting logits to -$\infty$ sets their likelihoods to 0.

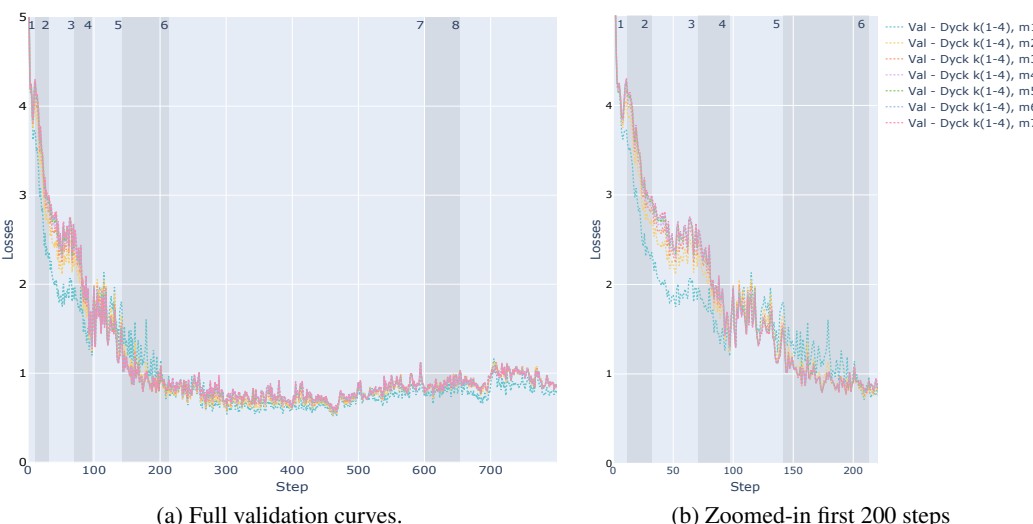

(a) Full validation curves.    (b) Zoomed-in first 200 steps

Figure 4: Normalized losses for Dyck languages of 1-4 parentheses types (vocab) and 1-7 depths.

The normalized losses for Dyck languages are shown in Figure 4. The inflection of losses continues through the third phase transition but disappears after phase 6. At phase 6, shallow expressions still exhibit higher loss values compared to lower-level expressions, which could suggest possible overfitting or memorization for certain lengths. Additionally, phase 8 does not show any distinguishable patterns in the normalized losses. This perhaps indicates that the effect of length can account for the drop in loss at phase 8, rather than syntactical generalization. However, the inflection of deeper expressions in phase 6 still persist, possibly suggesting a complex syntactical learning and generalization dynamics at various depths.

## 3.2 NORMALIZED PER-TOKEN LOSSES OF SYMBOLIC EXPRESSIONS

Additionally, we analyze the normalized per-token losses for a range of symbolic expressions, with a specific focus on the first two phase transitions that occur before the 100th training step. These transitions seem to mark significant points where key rules and foundational properties of FOL are learned. To gain a clearer understanding, we first review the specific rules incorporated into the training process, as summarized in Figure 5.

Several common patterns emerge across the various symbolic expressions. Notably, the parenthesis symbols "(" and ")" exhibit sharp, two-stage drops in loss values, corresponding directly to the first two phase transitions. These transitions are consistent with phases 2 and 4, as highlighted in Figure 2, and are observed across all expressions. This sharp reduction indicates that the model quickly grasps the hierarchical structure governed by these symbols in the early stages of learning.

In addition, various operators in first-order logic, such as "∧" and "∨," offer further insight into the process by which specific rules are learned. These operators appear to undergo a similar one-to-two-stage learning progression, though their transitions tend to occur slightly later, typically following the hierarchical acquisition of the parenthesis operators. The patterns exhibited by these operators shed light on the incremental and structured nature of learning in this context, reinforcing the idea that the model first internalizes the more basic structural elements before moving on to more complex logical operators.

We then examine the granular loss curves for complex rules that the model has not encountered before. Although our annotated complex test set for these unseen rules is limited, we still consider it a useful indicator of training dynamics. Figure 6 summarizes the findings, with the rule templates detailed in Appendix C. Notably, we see two staged phase transition with parenthetical operators. We also see drops in losses for other operators of FOL. However, beyond the second phase transition, we observe signs of memorization or overfitting in Figure 9 as the normalized losses begin to increase for these templated complex rule sets. Since these are limited, templated rules rather than inherent properties

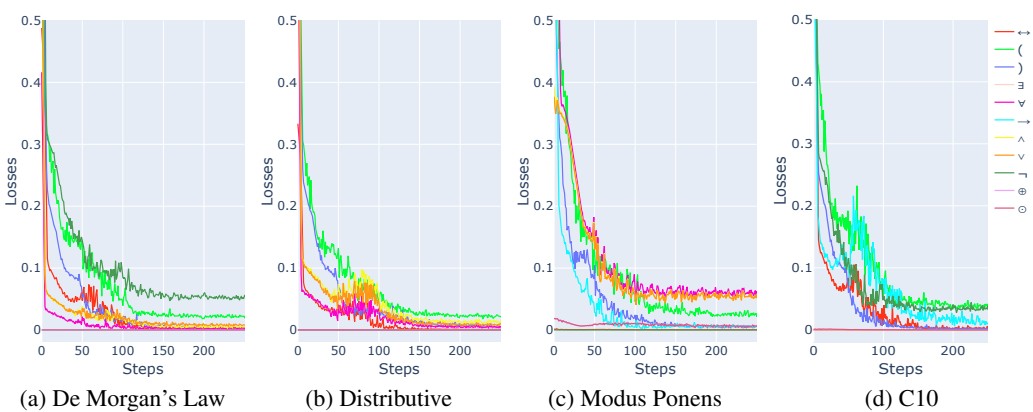

Figure 5: Per token losses of rules seen by the model during pretraining.

of FOL, this outcome may be expected. This abrupt increase in normalized losses also aligns with the third phase transition point shown in Figure 2, suggesting that the third phase transition may involve a trade-off between further generalization and memorization.

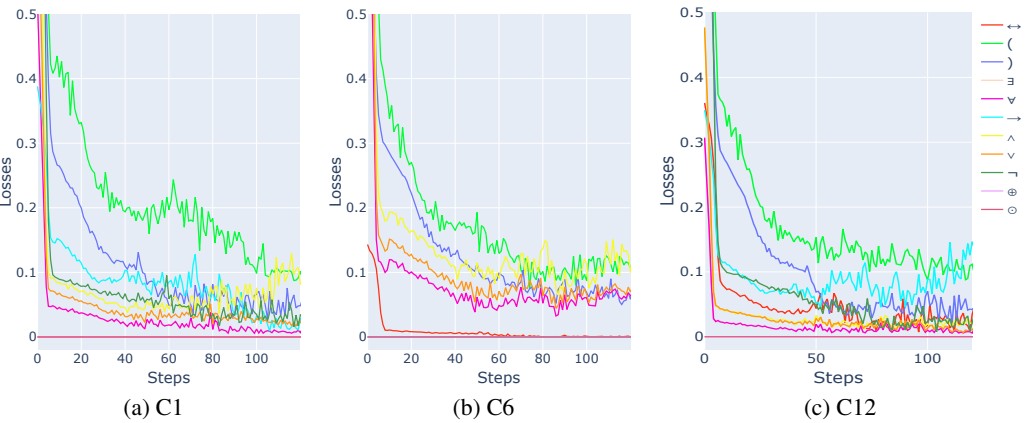

Figure 6: Per token losses of unseen rules.

### 3.3 EIGENVALUE ANALYSIS OF ATTENTION

A key aspect of first-order logic learning likely involves recognizing particular prior elements, determining their placement in a rule context and reproducing them in correct places, such as through prefix matching and copying. To gain further insight into the pretraining process, we consider the eigenvalues of attention matrices.

We follow the circuit formulation outlined by Elhage et al. (2021) and approximately define the QK and OV matrices as $W_Q^T W_K$ and $W_O W_V$, respectively, where $W_Q$, $W_K$, and $W_V$ represent the query, key, and value matrices of attention, and $W_O$ corresponds to the weights of the output linear layer. The QK-circuit describes the alignment between query and key values in the model, which can be interpreted as how much a key token's prediction relies on information from a query token. In contrast, the OV-circuit can be seen as a copying mechanism, transferring specific information to the resulting location. The eigenvalues of these matrices indicate how effectively the circuits scale an input vector. Large positive eigenvalues can be interpreted as a "copy score" for the OV circuit and a "prefix matching score" for the QK circuit. We track the eigenvalues of the attention matrices throughout pretraining, and the results are presented in Figure 7.

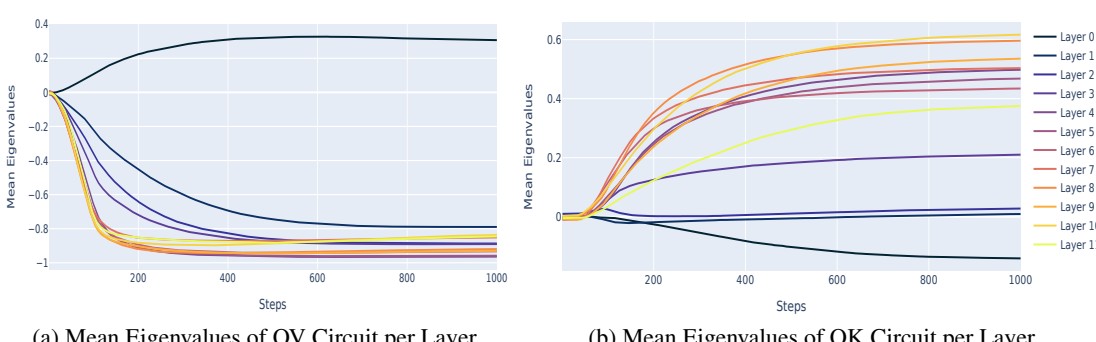

(a) Mean Eigenvalues of OV Circuit per Layer      (b) Mean Eigenvalues of QK Circuit per Layer

Figure 7: Traces of the Eigenvalues of Attention

The OV circuit appears to emerge within the first 100 steps of pretraining, indicating that the copying behavior is acquired around the time the basic inferencing of FOL are learned. In contrast, the prefix matching and QK eigenvalues continue to plateau well into the training, suggesting that focusing on occurring patterns and incorporating them into possibly more complex rules may be a more ongoing and challenging process. We also observe that the copying behavior appears to be concentrated in the first layer, while the prefix matching tends to occur in the deeper layers of the model. This could help clarify the transitions observed in the third and fourth hierarchical phase transitions in future work.

As highlighted by Olsson et al. (2022), transformers have a significant number of induction heads. Effectively copying relevant past context in the right places is essential for generating accurate expressions in FOL. This connection emphasizes the role of in-context learning in enhancing logical reasoning within transformer models.

## 4 CONTEXTUALIZING THE TRAINING TRAJECTORY AND COMPLEXITY: INSIGHTS AND FUTURE DIRECTIONS

We now consolidate our experimental findings to explain the training curve in Figure 2. Since first-order logic (FOL) is of higher complexity than Dyck languages, we expect that training on FOL should enable generalization to Dyck languages, even though the model has not been explicitly trained on them. Our results confirm this expectation, with the models exhibiting generalization at scale. These phases are marked by significant drops in the Dyck language losses, as illustrated in Figure 3. We observe that, at scale, this generalization unfolds in multiple phases, resembling a staircase pattern.

Empirically, we observe a flurry of activity during the first two phase transitions, both occurring before the 100th step. It appears that the model learns the fundamental properties and rules of FOL within these initial phases, as revealed by the fine-grained tracking of operator losses in Figure 5. Following this, the model starts to pick up on copying behavior in the 0th layer, signaled by the OV eigenvalues in Figure 7, which emerge shortly after the first 100 steps. The positivity of QK eigenvalues seem to develop more gradually in the later layers of the model, possibly indicating that prefix matching is learned well into the model training process.

The interpretation of the third hierarchical phase transition point, as well as the potential for a fourth transition, calls for further investigation. Notably, we observe an inversion in depth, where shallower expressions exhibit higher loss values than their deeper counterparts, as illustrated in both Figure 3 and normalized losses in Figure 4. Additionally, this phase transition point coincides with the point at which the trajectories of unseen rules in Figures 6 and 9 begin to display higher losses. Although our unseen test set is limited for this iteration of the study, we suspect that this may be due to the model overfitting or memorizing specific rules while generalizing on others. To address this, we need to evaluate the model on a much larger out-of-domain dataset, which is feasible in this context because FOL is a unique case where complexity can be meticulously annotated, including factors such as the number of variables, predicates, and depths of expressions.

Having tested on a lower-complexity out-of-domain set, we can now explore a higher-complexity out-of-domain set to examine whether we observe any phase transition behaviors. This could include more complex first-order logic sets or significantly simplified form of natural language reasoning sets. Such investigations will enhance our understanding of the role that complexity plays in phase transitions and pretraining.

Moreover, we can further explore pretraining in curriculum of varying complexity. While we do not delve deeper in this iteration, we also tried to "semantically prime" the model on the OpenWebText dataset (Gokaslan and Cohen, 2019) for a few hundred gradient iterations prior to the FOL pretraining, and the learning curves are shown in Appendix F. It seems to suggest that seeing structurally representative data at the beginning of training is crucial for generalization.

## 5 CONCLUSION

In this work, we explore the emergence of generalization at the scale of pretraining. While prior research has extensively studied grokking in small-scale models, our focus is on identifying similar dynamics at a much larger scale. We find that hierarchical generalization during pretraining follows staircase-like phase transitions. Furthermore, the acquisition of logically significant symbols and rules occurs at distinct stages throughout training. Although the pretraining loss and validation curves appears relatively smooth, multiple underlying learning and generalization processes are taking place at scale and at high data complexity. These findings suggest that we are only beginning to uncover the complexity of generalization in large models.

We are excited about the potential of this work to improve our understanding of how LLMs generalize during pretraining. While FOL seems abstract, it represents a formalized subset of natural language that captures key aspects of reasoning. Future work could help us understand how LLMs develop the ability to reason and the phases they undergo in this process, offering a useful analogy for reasoning in natural language. Additionally, this work provides a foundation for larger-scale interpretability on how phase transitions affect various model components, what is learned at each stage, where it occurs, and how learning is linked to the training data, with full transparency, thorough data annotation, as well as training granular checkpoints.

## 6 RELATED WORKS

**First-Order Logic (FOL) and Reasoning**    Propositional logic represents inferential relationships between true or false statements. Then, FOL extends it to represent far more complex relationships by introducing quantifiers (e.g. every as $\forall$), logical connectives (e.g. "and" as $\land$), and predicates (e.g. IsMadScientist(x)), allowing for a more expressive representation of knowledge. Then, by training an LLM on FOL, we can then examine how a model might learn logic and reasoning. We build upon some prior logic datasets such as LogicBench (Parmar et al., 2023), LogicNLI (Tian et al., 2021), and Folio (Han et al., 2022).

Beyond its syntactical representations, FOL may potentially be instrumental for probing how LLMs reason. Gulordava et al. (2018) argues that models can learn to track abstract hierarchical syntactic structure, even when they are unable to rely on semantic cues. However, recent work indicates that current language models are poorly skilled at basic boolean logic (Williams and Huckle, 2024). In parallel, some work shows that language models can be easily misled by simple patterns within the text such as lexical overlap (McCoy et al., 2019; Wu and Monz, 2023), entity boundary (Yang et al., 2023), word order (Zhang et al., 2023). Moreover, some work argues that LLMs lack true "undestanding" of logic (Yan et al., 2024), while others suggest that the current pretraining strategies cause models to replicate human reasoning patterns, including inherent biases. As with human cognition, one avenue for improving model reasoning is by teaching them to apply logic more effectively (Ozeki et al., 2024). Another study highlights the limitations in logical reasoning in today's LLMs by evaluating 25 models, showcasing instances of logically inconsistent judgments, even in advanced systems like GPT-4 (Holliday et al., 2024).

**Training Dynamics**    Previous research has investigated the dynamics of pretraining in language models, such as the study by Saphra and Lopez (2019), which examined how models implicitly encode linguistic features. Likewise, Choshen et al. (2021) and Evanson et al. (2023) observed

that linguistic generalizations are acquired in similar stages, regardless of the model's architecture, initialization, or data-shuffling methods. In masked language models, syntactic rules are acquired early (Chen et al., 2023), while world knowledge may emerge later and more unstably (Li et al., 2023; González and Nori, 2024). Notably, Olsson et al. (2022) observed that induction heads for in-context learning appear at key inflection points during pretraining. These findings hint at the emergence of generalized circuits at specific points during pretraining.

**Pretraining Curriculum**    There has been a long line of curiosity about the efficacy of curriculum learning for deep models Bengio et al. (2009). In particular relevance to this work, Wu et al. (2023) demonstrated a curriculum of nested boolean logic, gradating from simple to hard problems, which led to increased performance in logic learning. There are complex trade offs between memorization, forgetting and generalization throughout a model's training process. Chang et al. (2024) found that forgetting is influenced by factors like training data characteristics, batch size, and model size. Beyond the curriculum, these studies posit that de-duplication, large batch sizes, as well as paraphrasing are keys to better knowledge acquisition and retention.

**Generalization and Grokking**    Gromov (2023) introduced a sudden jump in generalization in a 2 layer neural network on a modular arithmetic task. This came to be known as grokking. Other works since have linked grokking to compression. Liu et al. (2022a) used a compression measure to track neural network evolution, and delayed memorization before generalization. Suggesting that grokking possibly occurs when models shift from relying on memorization and retrieval to discovering algorithms and heuristics which generalize better. The descent part of deep double descent—a phenomenon where test error initially decreases, then increases, and finally decreases again — seems illustrative of the competition between emerging memorization vs. generalization circuits within the model.

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

## A  FIRST ORDER LOGIC (FOL) CATEGORIES AND EXPLANATIONS

| FOL Inference Rule | Symbolic Expression | Explanation |
|---|---|---|
| Bidirectional Dilemma (BD) | $((p \rightarrow q) \wedge (r \rightarrow s)),$ $(p \vee \neg s) \models (q \vee \neg r)$ | If two conditional statements are true, given a true antecedent or a false consequent, the respective consequent is true or a respective antecedent is false. |
| Constructive Dilemma (CD) | $((p \rightarrow q) \wedge (r \rightarrow s)),$ $(p \vee r) \models (q \vee s)$ | If two conditional statements are true and at least one of their antecedents are true, then at least one of their consequents are true. |
| Destructive Dilemma (DD) | $((p \rightarrow q) \wedge (r \rightarrow s)),$ $(\neg q \vee \neg s) \models (\neg p \vee \neg r)$ | If two conditional statements are true, and one of their consequents has to be false, then one of their antecedents has to be false. |
| Disjunctive Syllogism (DS) | $((p \vee q) \wedge \neg p) \models q$ | Disjunctive elimination. If we know one of two statements, $p$ or $q$, to be true, and one of them is not true, the other must be true. |
| Hypothetical Syllogism (HS) | $((p \rightarrow q) \wedge (q \rightarrow r))$ $\models (p \rightarrow r)$ | Chain argument rule or transitivity of implication. |
| Modus Ponens (MP) | $((p \rightarrow q) \wedge p) \models q$ | Implication elimination rule. If $p$ implies $q$ and $p$ is true, the statement can be replaced with $q$. |
| Modus Tollens (MT) | $((p \rightarrow q) \wedge \neg q) \models \neg p$ | Implication elimination rule. If $p$ implies $q$ and $q$ is false, the statement can be replaced with *not* $p$. |
| Universal Instantiation (UI) | $\forall x P(x) \implies \exists a P(a)$ | If a statement $P$ holds for a variable $x$, then there exists a particular value $a$ for the statement to be true. |
| Existential Generalization (EG) | $\exists x P(x) \implies P(a)$ | If a statement $P$ holds true for some subset of variables $x$, then there's a particular value of $x = a$ for which $P$ holds true. |
| FOL proofs & general statements | - | - |

Table 2: First Order Logic (FOL) Inference Rule Categories and Explanations

| FOL Properties | Symbolic Expression |
|---|---|
| Distributive (Dist) | $(p \vee (q \wedge r)) \leftrightarrow ((p \vee q) \wedge (p \vee r))$ $(p \wedge (q \vee r)) \leftrightarrow ((p \wedge q) \vee (p \wedge r))$ |
| Association (AS) | $(p \vee (q \vee r)) \leftrightarrow ((p \vee q) \vee r)$ $(p \wedge (q \wedge r)) \leftrightarrow ((p \wedge q) \wedge r)$ |
| Tautology (TT) | $p \leftrightarrow (p \vee p)$ $p \leftrightarrow (p \wedge p)$ |
| Transposition (TS) | $(p \rightarrow q) \leftrightarrow (\neg q \rightarrow \neg p)$ |
| Importation (IM) | $(p \rightarrow (q \rightarrow r)) \leftrightarrow ((p \wedge q) \rightarrow r)$ |
| Exportation (EX) | $((p \wedge q) \rightarrow r) \rightarrow (p \rightarrow (q \rightarrow r))$ |
| Double Negation (DN) | $p \leftrightarrow \neg\neg p$ |
| De Morgan's Law (DM) | $\neg(p \wedge q) \leftrightarrow (\neg p \vee \neg q)$ $\neg(p \vee q) \leftrightarrow (\neg p \wedge \neg q)$ |
| Negation of XOR (NX) | $\neg(p \oplus q) \leftrightarrow (\neg p \oplus \neg q)$ $\neg(p \oplus q) \leftrightarrow (p \odot q)$ |
| Negation of XNOR (NN) | $\neg(p \odot q) \leftrightarrow (\neg p \odot \neg q)$ $\neg(p \odot q) \leftrightarrow (p \oplus q)$ |

Table 3: First Order Logic (FOL) Basic Properties

# B    ELIMINATION RULES

| | Symbolic Expression |
|---|---|
| E0 | $p \vee \text{True} \leftrightarrow \text{True}$ |
| E1 | $p \vee \text{False} \leftrightarrow p$ |
| E2 | $p \wedge \text{True} \leftrightarrow p$ |
| E3 | $p \wedge \text{False} \leftrightarrow \text{False}$ |
| E4 | $\text{True} \vee p \leftrightarrow \text{True}$ |
| E5 | $\text{False} \vee p \leftrightarrow p$ |
| E6 | $\text{True} \wedge p \leftrightarrow p$ |
| E7 | $\text{False} \wedge p \leftrightarrow \text{False}$ |
| E8 | $p \vee p \leftrightarrow p$ |
| E9 | $p \wedge p \leftrightarrow p$ |
| E10 | $p \wedge \neg p \leftrightarrow \text{False}$ |
| E11 | $p \vee \neg p \leftrightarrow \text{True}$ |
| E12 | $\neg p \wedge p \leftrightarrow \text{False}$ |
| E13 | $\neg p \vee p \leftrightarrow \text{True}$ |
| E14 | $p \wedge (p \vee q) \leftrightarrow p$ |
| E15 | $p \wedge (\neg p \vee q) \leftrightarrow (p \wedge \neg p) \vee (p \wedge q)$ 
 $\leftrightarrow \text{False} \vee (p \wedge q) \leftrightarrow p \wedge q$ |
| E16 | $p \wedge (\neg p \vee q) \leftrightarrow \text{False} \vee (p \wedge q) \leftrightarrow p \wedge q$ |
| E17 | $p \wedge (\neg p \vee q) \leftrightarrow (p \wedge \neg p) \vee (p \wedge q) \leftrightarrow p \wedge q$ |
| E18 | $p \vee (p \wedge q) \leftrightarrow p$ |
| E19 | $p \vee (p \wedge q \wedge r) \leftrightarrow p$ |
| E20 | $r \vee (p \wedge q \wedge r) \leftrightarrow r$ |
| E21 | $r \vee (p \wedge q \wedge r \wedge s) \leftrightarrow r$ |
| E22 | $p \vee (\neg p \wedge q) \leftrightarrow (p \vee \neg p) \wedge (p \vee q)$ 
 $\leftrightarrow \text{True} \wedge (p \vee q) \leftrightarrow (p \vee q)$ |
| E23 | $p \vee (\neg p \wedge q) \leftrightarrow \text{True} \wedge (p \vee q) \leftrightarrow (p \vee q)$ |
| E24 | $p \vee (\neg p \wedge q) \leftrightarrow (p \vee \neg p) \wedge (p \vee q) \leftrightarrow (p \vee q)$ |
| E25 | $p \vee \neg (p \wedge q) \leftrightarrow p \vee (\neg p \vee \neg q)$ 
 $\leftrightarrow (p \vee \neg p) \vee \neg q \leftrightarrow \text{True} \vee \neg q \leftrightarrow \text{True}$ |
| E26 | $p \vee \neg (p \wedge q) \leftrightarrow p \vee (\neg p \vee \neg q)$ 
 $\leftrightarrow p \vee \neg p \vee \neg q \leftrightarrow \text{True} \vee \neg q \leftrightarrow \text{True}$ |
| E27 | $p \vee \neg (p \wedge q) \leftrightarrow (p \vee \neg p) \vee \neg q \leftrightarrow \text{True} \vee \neg q \leftrightarrow \text{True}$ |
| E28 | $p \vee \neg (p \wedge q) \leftrightarrow p \vee (\neg p \vee \neg q) \leftrightarrow \text{True} \vee \neg q \leftrightarrow \text{True}$ |
| E29 | $p \vee \neg (p \wedge q) \leftrightarrow p \vee (\neg p \vee \neg q) \leftrightarrow (p \vee \neg p) \vee \neg q \leftrightarrow \text{True}$ |
| E30 | $p \wedge \neg (p \vee q) \leftrightarrow p \wedge (\neg p \wedge \neg q) \leftrightarrow (p \wedge \neg p) \wedge \neg q$ 
 $\leftrightarrow \text{False} \wedge \neg q \leftrightarrow \text{False}$ |
| E31 | $p \wedge \neg (p \vee q) \leftrightarrow (p \wedge \neg p) \wedge \neg q \leftrightarrow \text{False} \wedge \neg q \leftrightarrow \text{False}$ |
| E32 | $p \wedge \neg (p \vee q) \leftrightarrow p \wedge (\neg p \wedge \neg q) \leftrightarrow \text{False} \wedge \neg q \leftrightarrow \text{False}$ |
| E33 | $p \wedge \neg (p \vee q) \leftrightarrow p \wedge (\neg p \wedge \neg q) \leftrightarrow (p \wedge \neg p) \wedge \neg q \leftrightarrow \text{False}$ |

Table 4: First Order Logic (FOL) Elimination Rules

## C  COMPLEX FOL EXPRESSIONS

| | Symbolic Expression | Combination of FOL Rules | Included in Training |
|---|---|---|---|
| C1 | $(((a \vee b) \to q) \wedge \neg q) \to (\neg a \wedge \neg b)$ | MT + DM | ✗ |
| C2 | $(((a \wedge \neg b) \to q) \wedge \neg q) \to (\neg a \vee b)$ | MT + DM + DN | ✓ |
| C3 | $(p \to q), (q \to r), (s \to t), (\neg t \vee \neg r) \to (\neg p \vee \neg s)$ | TS + DD | ✓ |
| C4 | $(p \vee (q \wedge (a \vee b)))$ $\leftrightarrow ((p \vee q) \wedge ((p \vee a) \vee b))$ | DS + AS | ✓ |
| C5 | $(p \wedge ((a \wedge b) \vee q \vee r))$ $\leftrightarrow (((p \wedge a) \wedge b) \vee (p \wedge q) \vee (p \vee r))$ $\leftrightarrow (((p \wedge a) \wedge b) \vee (p \wedge (q \vee r)))$ | DS + AS | ✓ |
| C6 | $((p \wedge q \wedge r) \vee (a \wedge p \wedge b) \vee (c \wedge d \wedge e))$ $\leftrightarrow ((p \wedge ((q \wedge r) \vee (a \wedge b))) \vee (c \wedge d \wedge e))$ | DS + AS | ✗ |
| C7 | $(p \vee (q \wedge r \wedge (a \vee b) \wedge s))$ $\leftrightarrow ((p \vee q) \wedge (p \vee r) \wedge (p \vee a \vee b) \wedge (p \vee s))$ | DS + AS | ✓ |
| C8 | $(p \vee (q \wedge (p \vee b) \wedge r))$ $\leftrightarrow ((p \vee q) \wedge (p \vee b) \wedge (p \vee r))$ $\leftrightarrow (p \vee (q \wedge b \wedge r))$ | DS + AS + TT | ✓ |
| C9 | $\neg(p \vee (q \wedge (\neg a \vee b) \wedge \neg r))$ $\leftrightarrow \neg((p \vee q) \wedge (p \vee \neg a \vee b) \wedge (p \vee \neg r))$ $\leftrightarrow (\neg(p \vee q) \vee \neg(p \vee \neg a \vee b) \vee \neg(p \vee \neg r))$ $\leftrightarrow ((\neg p \wedge \neg q) \vee (\neg p \wedge a \wedge \neg b) \vee (\neg p \wedge r))$ $\leftrightarrow (\neg p \wedge (\neg q \vee (a \wedge \neg b) \vee r))$ | DS + DM + DN | ✗ |
| C10 | $(\neg p \to q) \leftrightarrow (\neg q \to p)$ | TS + DN | ✓ |
| C11 | $(p \to \neg q) \leftrightarrow (q \to \neg p)$ | TS + DN | ✓ |
| C12 | $((a \wedge b) \to q) \leftrightarrow (\neg q \to (\neg a \vee \neg b))$ | TS + DM | ✗ |
| C13 | $(p \to (\neg a \vee \neg b)) \leftrightarrow ((a \wedge b) \to \neg p)$ | TS + DM | ✓ |
| C14 | $\neg((a \vee b) \oplus c \oplus d) \leftrightarrow (\neg(a \vee b) \oplus \neg c \oplus \neg d)$ | DM + NX | ✓ |
| C15 | $\neg(c \oplus (\neg a \vee b) \oplus d) \leftrightarrow (\neg c \oplus (a \wedge \neg b) \oplus \neg d)$ | DM + NX | ✗ |
| C17 | $\neg(p \odot q \odot (a \vee \neg b)) \leftrightarrow (\neg p \odot \neg q \odot (\neg a \wedge b))$ | DM + NN | ✓ |
| C18 | $((a \wedge b) \to q), ((a \wedge \neg c) \to s), (\neg q \vee \neg s)$ $\to ((\neg a \vee \neg b) \vee (\neg a \vee c))$ $\to (\neg a \vee \neg b \vee c)$ $\to \neg(a \wedge b \wedge \neg c)$ | DD + DN + DM + AS + TT | ✗ |
| C20 | $((a \vee b) \to q), (r \to s), (a \vee b \vee r) \to (q \vee s)$ | CD + AS | ✓ |
| C21 | $(p \to (a \vee b)), (r \to s), (p \vee r) \to (a \vee (b \vee s))$ | CD + AS | ✓ |
| C22 | $(p \to q), ((a \vee \neg b) \to s), (p \vee \neg s)$ $\to (q \vee (\neg a \wedge b))$ $\to ((q \vee \neg a) \wedge (q \vee b))$ | BD + DN + DM + DS | ✗ |
| C23 | $(p \to q), ((\neg a \wedge \neg b) \to s), (p \vee \neg s)$ $\to q \vee \neg(\neg a \wedge \neg b)$ $\to (q \vee a) \vee b$ | BD + DM + AS | ✓ |

Table 5: Complex FOL Expressions. (BD = Bidirectional Dilemma, CD = Constructive Dilemma, DD = Destructive Dliemma, MT = Modus Tollens, DM = De Morgan's, DN = Double Negation, DS = Distribution, AS = Association, TS = Transposition, TT = Tautology, NN = Negation of XNOR, NX = Negation of XOR)

# D TRAINING DATA

| | # Examples | # Tokens | | # Examples | # Tokens |
|---|---|---|---|---|---|
| BD | 102.43K | 778.57K | DM | 740.92K | 34.11M |
| CD | 856.78K | 71.70M | Dist | 268.76K | 17.97M |
| DD | 806.15K | 71.70M | XOR* | 17.46K | 793.75K |
| DS | 321.31K | 12.79M | XNOR* | 15.42K | 668.07K |
| HS | 429.94K | 23.20M | XOR-XNOR* | 14.49K | 577.84K |
| MP | 237.80K | 7.49M | | | |
| MT | 285.19K | 11.10M | | | |
| UI | 123.00K | 4.03M | | | |
| EG | 9.10K | 263.0K | | | |
| General/fol proof | 2.27M | 286.12M | | | |
| C2 | 94.36K | 5.26M | E0 | 5.23K | 83.92K |
| C3 | 108.56K | 20.21M | E1 | 5.34K | 141.32K |
| C4 | 102.73K | 8.03M | E2 | 5.28K | 147.74K |
| C5 | 107.64K | 16.67M | E3 | 5.46K | 88.60K |
| C7 | 109.03K | 17.57M | E4 | 5.18K | 82.91K |
| C8 | 97.90K | 12.38M | E5 | 5.31K | 146.08K |
| C10 | 102.43K | 3.84M | E6 | 5.35K | 148.27K |
| C11 | 86.92K | 3.55M | E7 | 5.28K | 84.61K |
| C13 | 93.22K | 5.17M | E8 | 5.40K | 188.11K |
| C14 | 109.20K | 10.03M | E9 | 5.37K | 194.38K |
| C17 | 108.53K | 10.04M | E10 | 5.23K | 140.78K |
| C20 | 109.10K | 10.58M | E11 | 5.22K | 139.83K |
| C21 | 109.37K | 11.21M | E12 | 5.16K | 134.20K |
| C23 | 109.45K | 14.94M | E13 | 5.18K | 135.07K |
| | | | E14 | 5.60K | 135.07K |
| | | | E15 | 5.58K | 720.92K |
| | | | E16 | 5.56K | 441.38K |
| | | | E17 | 5.58K | 569.09K |
| | | | E18 | 5.61K | 256.89K |
| | | | E19 | 5.37K | 291.75K |
| | | | E20 | 5.21K | 286.32K |
| | | | E21 | 5.37K | 352.57K |
| | | | E22 | 5.61K | 734.95K |
| | | | E23 | 5.59K | 470.14K |
| | | | E24 | 5.63K | 586.41K |
| | | | E25 | 5.58K | 703.58K |
| | | | E26 | 5.57K | 714.06K |
| | | | E27 | 5.58K | 501.50K |
| | | | E28 | 5.57K | 502.50K |
| | | | E29 | 5.59K | 625.27K |
| | | | E30 | 5.59K | 720.77K |
| | | | E31 | 5.59K | 499.31K |
| | | | E32 | 5.59K | 498.49K |
| | | | E33 | 5.60K | 648.88K |
| **Complex Total** | **1.45M** | **149.48M** | **Eliminations Total** | **185.09K** | **12.24M** |
| Random | 15.26M | 1.89B | | | |
| **Total** | **24.07M** | **2.67B** | | | |

Table 6: Full Breakdown of the Training Dataset. The labels are consistent with the FOL types described in Table 2, Table 3, Table 4, and Table 5. Note: not all basic properties (Table 3) of FOL were included explicitly in generation. This is because we qualitatively saw that the massive random generation sufficiently and implicitly (and sometimes explicitly) captured the basic properties.

*We explicitly included negations of XOR, negations of XNOR and the equivalences between XOR and XNOR.

# E NORMALIZED PER TOKEN LOSS FOR COMPLEX RULES

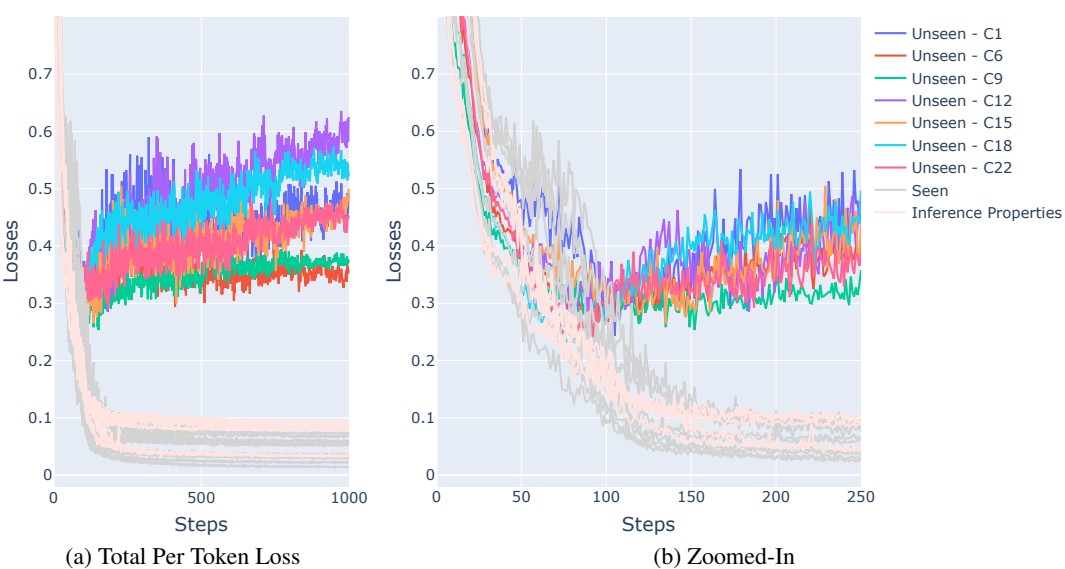

(a) Total Per Token Loss

(b) Zoomed-In

Figure 8: Total Per token losses.

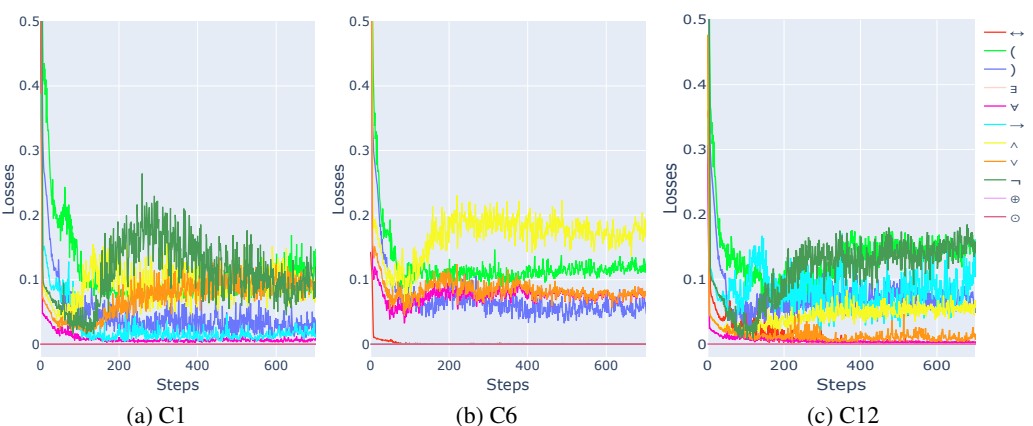

(a) C1

(b) C6

(c) C12

Figure 9: Per token losses of unseen rules.

# F  SEMANTICALLY PRIME, THEN PRETRAIN

We then experiment with semantically priming the model on natural language first to see how it affects the representations and model performance. We prime the model on OpenWebText for the first few hundred iterations. During each iteration, the model is estimated to see 491,520 tokens.

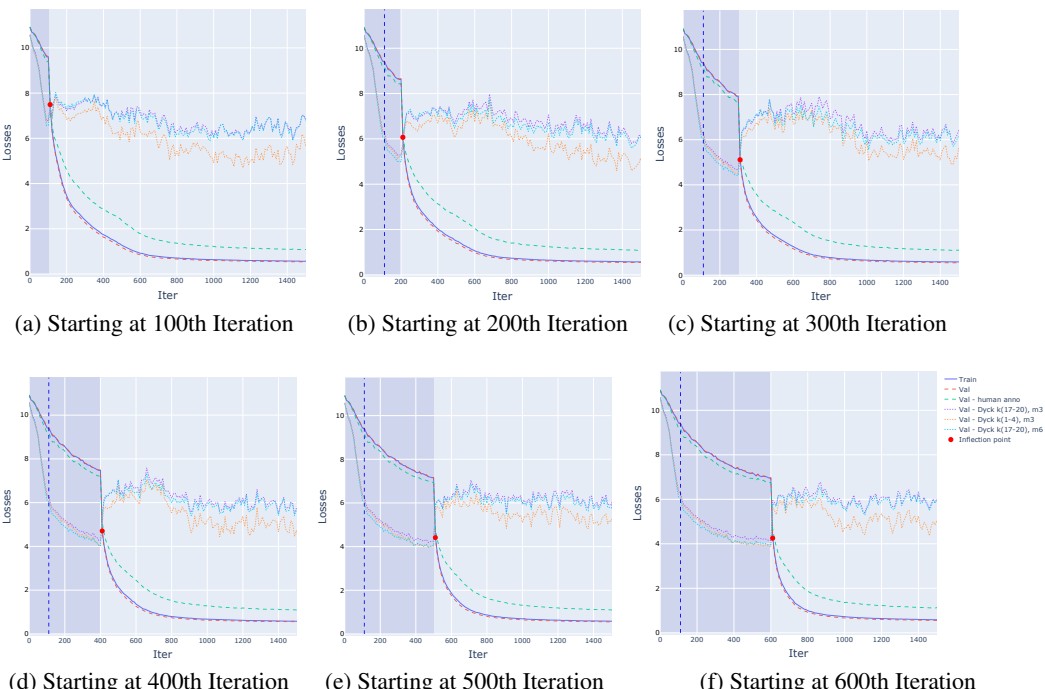

(a) Starting at 100th Iteration  (b) Starting at 200th Iteration  (c) Starting at 300th Iteration

(d) Starting at 400th Iteration  (e) Starting at 500th Iteration  (f) Starting at 600th Iteration

Figure 10: Training curves for semantically primed models. The shaded blue regions represent semantic priming on OpenWebText.

The results are illustrated in Figure 10. After the first 100 steps of semantic priming, the generalization curves for Dyck languages fail to reach the same low loss levels, suggesting that semantic priming disrupts phase transitions. Many structural generalizations seem to occur within the first 200 iterations, indicating that semantic priming has a detrimental effect on generalization. This could explain why fine-tuning in some cases yields only limited improvements when (structurally) similar data were not part of the pretraining stage. A potential follow-up experiment would be to incorporate FOL data into the web priming dataset and compare the outcomes. We may also experiment with hyperparameters such as learning rate matching and drop out.

