# OpenReview forum: "(Pre-)training Dynamics: Scaling Generalization with First-Order Logic"
_ICLR.cc/2025/Conference — ICLR 2025 Conference Withdrawn Submission_

### Official Review · Reviewer_EUcb · 2024-10-30

**Soundness:** 1
**Presentation:** 2
**Contribution:** 1
**Rating:** 3
**Confidence:** 4

**Summary:**

The paper develops a first-order logic (FOL) dataset by combining synthetic data generation with data generated using large language models based on existing FOL-based tasks.
It uses this dataset as a simplified setting to study the training dynamics of a 125M parameter language model and uses Dyck languages and held-out first-order logic expressions to empirically study different types of generalization.

**Strengths:**

The generation of complex synthetic datasets as a formalized subset of natural language is a potentially useful tool for understanding the training dynamics and systematically evaluating the shortcomings of language models on smaller scales.

**Weaknesses:**

The presentation of the empirical results lacks clarity and it is unclear to me what the concrete findings and especially their implications are (see questions).
In particular, it is not clear to me why it is interesting to test for generalization on a task (Dyck languages) that is a subset of the training task (first-order logic) and the generality of the observed empirical phenomena across various experimental conditions has as far as I understand not been systematically investigated (see questions).

**Questions:**

> Around 70% of the training data consists of the randomly generated and guaranteed correct expressions and their equivalent simplifications.

- What does the LLM-generated data bring on top of purely using symbolically generated data in SymPy. The latter gives verifiable control over the data distribution for testing systematic generalization.

> Existing human curated datasets such as Folio and LogicBench were used as another “human validation set.”

- If Folio and LogicBench are used to generate data as in-context examples, there might be data leakage into the validation set. Do you control for this?

> Having tested on a lower-complexity out-of-domain set, we can now explore a higher-complexity out-of-domain set to examine whether we observe any phase transition behaviors. This could include more complex first-order logic sets or significantly simplified form of natural language reasoning sets. Such investigations will enhance our understanding of the role that complexity plays in phase transitions and pretraining.

- How do you justify the choice of Dyck languages to test generalization. They are a arguably simple subset of FOL and thus a prerequisite for being able to truly solve FOL expressions.

> Sympy was used to generate 400-500K syntactical rules, it is highly unlikely to have generated our exact sets of complex rules, with the same variables, predicates, and orders of operations.

- Would it be possible to only include syntactical rules that are verifiably not covered by the train set? In general, the analysis of generalization to held-out FOL tasks is potentially more interesting since it allows to test for systematic/compositional generalization to novel compositions of seen FOL expressions.

- To what extent are the phase transitions you identify stable across experimental settings (e.g. seed, ordering of the data, model size, optimization hyperparameters, ...)? Such an analysis would help to more convincingly make the case that the transitions you identify aren't just random fluctuations due to the particularities of the experiments.
- What are the concrete implications of identifying such phase transitions for our understanding of the pretraining dynamics?

- Would it be possible to report a more interpretable notion of correctness other than the loss such as accuracy of masked FOL predictions to get a better sense of performance?

- Related to the previous point, how does the task difficulty compare to other algorithmic datasets? Do the loss values achieved by the 250M model here mean the model can perfectly solve FOL tasks at this scale?

- Why is FOL considered "semi-algorithmic". In what sense is it different from algorithmic tasks? In my understanding FOL statements can be algorithmically evaluated which is demonstrated by the use of sympy to algorithmically generate the data.


## Minor

> Sympy relies on graphical representation of FOL operations

- This statement sounds strange to me. Is this a typo and you meant "symbolic representation" or what are you referring to here?

- I am assuming you are using a causal next-token prediction cross-entropy loss but this is not stated in the manuscript.

- I think conventionally loss plots should be reported in log scale

---

### Official Review · Reviewer_jRVX · 2024-11-02

**Soundness:** 1
**Presentation:** 2
**Contribution:** 1
**Rating:** 3
**Confidence:** 3

**Summary:**

The paper studies generalisation in transformer models, using a benchmark based on first-order logic. First, authors prepare a novel dataset consisting of FOL statements: first, generating a set of FOL tautologies synthetically, and then replacing the abstract predicates and variables with natural-language instantiations (such as replacing $P(x) \vee \neg P(x)$ with "$married(man)\ \vee \neg married(man)$"), using off-the-shelf LLMs.

Then, the paper describes training a GPT-2-based LLM on those statements, while investigating the performance across different validations sets. By inspecting the training and validation curves, as well as additional statistics of the model behavior (mean eigenvalues of the $OV$ and $QK$ matrices), authors attempt to gain further insight into the phase transitions of the model corresponding to different stages of generalisation. The main finding is claimed to be that generalisation for different types of problems (different types of formulae) happens progressively, in a stair-case like type of pattern, with sharp improvements separated by periods of relatively flat performance. In addition, the order in which different types of generalisation abilities are developed (such as being able to track the parenthesis, or understand conjunction and disjunction) tracks the perceived complexity of those problems.

**Strengths:**

- The paper focuses on an important and relevant problem of generalisation in LLMs.
- Using first-order logic seems like a reasonable middle ground between simple models of computation, and natural language in its full generality.
- The dataset generated for the paper might be a valuable asset for the ML community: statements in FOL were generated synthetically and proved to be valid, and then instantiated in a natural language by off-the-shelf LLMs.

**Weaknesses:**

- The writing, although reasonably coherent, does not tell a clear story, and reveals the rudimentary and provisional nature of the paper's central findings. After describing the initial setup, the paper consists of a few sections of various qualitative observations, explanations and guesses about the training dynamics. Examples are numerous:
	 - "we hypothesize that lower depth expressions in phase 4 and beyond exhibit higher loss because the model has fewer previous tokens to conditon on"
	 - "we suspect that longer expressions my help the model narrow its distribution to the valid tokens"
	 - "at phase 6, shallow expresssions still exhibit higher loss [...], which could suggest possible overfitting or memorisation for certain lengths", etc.
- All those claims:
	 - are purely post-hoc, i.e. in no way pre-registered (I'm not requiring formal pre-registration here - just on the level of some a priori theory or common sense suggesting they should happen)
	 - rely on authors visually inspecting noisy training curves
	 - carry no attempt to further formally investigate them, suggest alternative hypotheses, repeat experiments (for different hyperparameters, optimiser settings or initialisations), form a unifying theory, or make predictions for the future
	 - are at the same time vague (as to what really happened) and very specific (as to pertaining only to that specific model and training run)
 - Both the form and the substance thus make it difficult to get a clear conclusion from the experiment.
 - There is no attempt to formally explain or further explore the difficulty of learning different types of FOL rules (e.g. from the point of view of computational complexity)
 - The central connection to grokking is dubious. The original Power et al. paper showed that training has to continue for up to 1000x times longer than until convergence, for a very small model and a very simple, purely synthetic task. Here, authors use GPT-2-small sized model, and train it on 10,000 iterations - comparing it to the implementation in the nanoGPT repository, the model is trained for 600 000 iterations by default.
 - Apart from the dataset contribution, at the current shape, the paper can be taken as a case study and a point of comparison for future work.

**Questions:**

Questions:
- What are the central insights from training this model, and what concrete predictions, recommendations, or other contributions does it make for future experiments?
- Why do the authors think that the experiments and results will replicate under other datasets, architectures, hyperparameters and optimisers?
- Why are 10 000 training iterations enough to induce grokking here?
 - Which of the claims in the paper were expected a prior, and which weren't? What theory did it attempt to confirm or refute?
 - Why not start with a propositional logic, as the next step after Dyck languages?


Other comments:
 - In the appendix A, the Universal Instatiation rule is false - it's not only a typo, but the explanation is also false. The correct rule is $\forall_x P(x) \implies P[x/t]$ for all terms $t$.
 - In the discussion, the first paragraph on the distinction between propositional and first-order logic is false. Propositional logic includes all logical connectives. First-order logic adds predicates and quantifiers.
 - It would be interesting to include discussion on the limits of the transformer-based logic reasoning, similarly to how Yao et al. do it for limited-depth Dyck languages.

---

### Official Review · Reviewer_9NdB · 2024-11-03

**Soundness:** 1
**Presentation:** 2
**Contribution:** 3
**Rating:** 3
**Confidence:** 4

**Summary:**

This work considers the pre-training of a transformer on a first-order logic (FOL) task. This FOL tasks is new and presented as an intermediary between natural language (the ultimate use case of transformers) and more simple languages such as Dyck grammars. The ability of the model to train on FOL and generalize to Dyck grammars is considered. Moreover human created datasets as also used as validation sets to complement the new dataset which is  generated from large language models.

**Strengths:**

# Originality
To my knowledge transformers have not been tested in a similar setting yet at this scale - an explicit FOL task with 400k to 500k syntactical rules. This lends insight into how transformers generalize and form hierarchical computation. The use of multiple generalization tests such as the human reasoning datasets and Dyck grammars help the work stand out too.

# Quality
The use of a number of tests of generalization is appreciated and supports the quality of the experimental design. Exploring Dyck grammars and ensuring the  model does not lose simple reasoning in learning FOL is an interesting and clever test. In addition the experimental design is appropriate for exploring the generalization dynamics of transformers. Indeed, the clear link between FOL as an intermediate step towards understanding transformers on natural language is one of the  biggest strengths of this work. Finally, the additional analysis which is conducted is useful, such as the eigenvalue analysis of the attention matrices. I also appreciate a couple subtle but important details such as the verifiability of the Dyck grammars and FOL task being of key benefit. This all supports this work as having quite high quality experimental design which is a great strength.

# Clarity
Overall the paper is well written and easy to follow. The paper is structured intuitively which aids the readability. The project is well motivated and place within the broader literature which helps understand the work and its motivations.

# Significance
Obtaining a clear understanding of how transformers learn FOL is a clear advance and would likely lead to a wealth of new work. The new large dataset presented could also find use in future work. Lastly, grokking remains a mysterious phenomenon in training neural networks and this paper proposes to shed light on its emergence as well.  This further supports the significance of the research

**Weaknesses:**

# Clarity
As  noted under strengths I think the experimental design of this work is probably its largest strength. However, in some places the description of the tasks and datasets are left quite vague. For example in the FOL task, how much of the text is shown? I assume it is everything up to the final $\rightarrow$ or similar structure (so that the model has to determine the truth or appropriate conclusion given the predicates) but it is never actually said. Similarly for the Dyck datasets, how is the model actually presented with input here and what must it output?

My second critique on the clarity is the figures. Firstly the font size is quite small, but more importantly, in some cases it is very difficult to distinguish between the lines. For example, does Figure 3a) have two blue dotted lines? Considering that the blue line is one of the most important ones for the result, this lack of clarity is a hinderance. In most cases the vague captions do little to help improve the clarity of the figures.  I suggest the figures be made much clearer.

Two minor points: on the readability of the examples in Table 1. I suggest spaces be added to make these examples more readable. For example with $\forall\ x\ \text{AttendParty}(x)$ (note the spacing) rather than $\forall x \text{AttendParty}(x)$. On lines 180 to 183 the statement "Furthermore, we create...test of generalization". This statement could be made clearer as it comes immediately after the Dyck grammars description (which again could use more detail on what exactly the transformer is seeing and doing). So it was a bit of a sudden change of topic.

# Quality
While the experimental design is good, I have some concerns on other aspects of this work. My first concern is that the novel dataset is generated from LLMs. While I do not disagree with this on a fundamental level, little concern is given to the unreliable nature of this approach. There are many works which use generated data to help learning, but this is slightly different as in this case it is presented as ground truth and not scrutinized at all. I think this is still necessary. I do, however, feel this is mitigated by the human labelled dataset and note that in Figure 2 the performance is similar between this validation set and the generated set. The lack of consideration here still seems to be against quality to a degree.

My primary critique, however, is that I just don't see most of the results the paper claims to obtain. Firstly, it is never described how a learning transition is defined. The numbers at the top of the loss curves are not even explained in the figure captions. The biggest offender of this is phase 6: what about epoch 140 (roughly) is important on this curve? At least for the others there is something like a peak or trough nearby (even if it is debatable that this means anything like in phase 8). But phase 6 to me corresponds to no clear salient point in the loss trajectory in Figure 2. My understanding is that these phases are determined based on Figure 2, so I can ignore the lack of key points in the other figures (although this could be clearer too if I am correct), but there should be no doubt in Figure 2 what I should be seeing.

Relatedly, I am not certain the inflection point in Figure 3 corresponds to something very meaningful. This is not helped by the difficulty of separating lines but it looks like at least two trajectories are following the same path with the blue line in Figure 3a. Figure 3b then seems to remove some lines. But either way, I don't think you can conclude that the losses of shallower expression increase compared to deeper expression based off of one setting. All depths seems to follow the exact same trajectory except for m1which still seems  to track very closely. Without a notion of variance for these trajectories it is tough to claim much of anything from a gap of that size which appears for only on task setting. Figure 4 has similar claims made about shallow expression exhibiting higher loss but all of these lines look overlapping to me by phase 6.

My final large concern is with the use of the Dyck grammars - again I think it is possible that the exact setup wasn't explained and I may be assuming something incorrectly - but it seems quite unreasonable to expect the model to generalize to these datasets. Evaluating FOL statements seems fundamentally different to me that evaluating the Dyck grammars. This is supported by the fact that in Figure 2 the Dyck grammars perform quite poorly. In the first paragraph on Section 4 it is then claimed that this experiment demonstrates that the model generalizes at scale. Given the relatively high loss, this seems premature.

Some smaller concerns:
- It is unclear to me how any of the results in this work relate to grokking. I'm not certain why this is brought up as none of these models grok. The connection in the "Generalization and Grokking" paragraph of the related work does not strike me as a necessary link, given the  results of this work.
- Similarly the relation to double descent doesn't make sense. However this may be due to a misunderstanding. Double descent does not relate to time but rather the number of parameter in the model. So describing the phenomenon as period changes in test error, as is done on line 507, is incorrect. The x-axis for double descent is not time or epochs, it is number of model parameters.
- In Figure 7a) it is stated that copying behaviour only emerges in the first layer, but if you only consider magnitude then later layers clearly have non-negligible copying behaviour. Why are negative eigenvalues ignored here?
- For Figure 5: I cannot see how it can be claimed that the brackets are learned before any other symbol. As far as I can tell all symbols begin learning immediately and converge at similar times. The link then to phase transitions in the loss of Figure 2 seems even more dubious.
- Figure 6: this figure seems to follow completely different dynamics to the Figure 5 losses, which makes me more concerned about the connection to phases in Figure 2 - as those phases are mainly based on test data (I believe this from Line 205 - again it isn't explained fully).

**Questions:**

I have grouped in my questions above where they emerged naturally. I would appreciate if they were answered, and have no further questions here.

---

### Official Review · Reviewer_PPXK · 2024-11-05

**Soundness:** 3
**Presentation:** 2
**Contribution:** 2
**Rating:** 5
**Confidence:** 4

**Summary:**

This paper studies the pre-training dynamics in transformer models to examine if observations from simple algorithmic tasks, such as grokking, transfer to larger, more realistic setups. To achieve this, the authors designed a synthetic dataset using first-order logic (FOL) with 2.5 billion tokens and trained a GPT-2 model with 125 million parameters, a setup larger than typical grokking studies. Using this setup, the authors analyzed how logical reasoning capabilities emerge and observed multiple distinct "phase transitions" in the pre-training dynamics, where the model learns increasingly complex operators and rule sets within FOL. Finally, the authors suggest that part of these observations can be explained by the early formation of the OV circuit before the formation of the QK circuit.

**Strengths:**

* The approach of using a synthetic pre-training dataset of FOL is timely in testing whether observations made in simple studies can translate to a larger scale.
* I appreciate the efforts to conduct the synthetic study with a scale larger than a typical study in mechanistic interpretability.

**Weaknesses:**

I am on board with the overall aspiration for this study to bridge the gap, overcoming the trade-off that "small scale study sacrificing realism to offer mechanistic insights" and "larger scale study gaining realism but lacking mechanistic insights". However, the presented results at intermediate scale and complexity, where majority of claims are "suspicions" rather than confirmation of falsifiable hypotheses, seems to lack neither of practicality nor clear mechanistic hypothesis and testing of it. Furthermore, the writing and presentation of figures requires significant improvements to meet the standard of ICLR conference. More concretely, for example, main result figures, i.e., Figures 2-6 are all many lines of loss curves without any explanation in the caption beyond a single sentence. Also, given its scientific nature of this paper, in its paper writing, the hypotheses need to be formulated more carefully so that they're cleanly testable.

**Questions:**

* Main result figures, Figures 2-6 are loss curves with shaded regions highlighting distinct "phases". Could you present any formal definition of those "phases", or did you just draw shades by using your subjective sense looking at the loss curves with your eyes?
* How does the main take away of "hierarchical generalization" relate to recent works that make similar claims in study of Transformers pre-trained on synthetic data of formal language, e.g., PCF(S)G?
* Given that FOL has a well-defined formal structure, have you considered analyzing the learned representations through the lens of formal grammar theory?

**Details Of Ethics Concerns:**

Does not apply.

---

### Note · Authors · 2024-11-13

**Comment:**

Hi all, thanks for all the feedback! Our work is rather preliminary at this stage, but decided to submit in order to gather feedback to see if our project indeed 1) addresses an existing gap, 2) proposes novel directions, and 3) what problems the community might have with our work/direction. One thing we do want to note is, this was a study on generalization at scale, not specifically on grokking. We'll make sure to clarify that in the continuation. Thanks again!

**Withdrawal Confirmation:**

I have read and agree with the venue's withdrawal policy on behalf of myself and my co-authors.